# Age-Related Dysfunction in Proteostasis and Cellular Quality Control in the Development of Sarcopenia

**DOI:** 10.3390/cells12020249

**Published:** 2023-01-07

**Authors:** Hector G. Paez, Christopher R. Pitzer, Stephen E. Alway

**Affiliations:** 1Department of Physiology, College of Medicine, University of Tennessee Health Science Center, Memphis, TN 38163, USA; 2Integrated Biomedical Sciences Graduate Program, College of Graduate Health Sciences, University of Tennessee Health Science Center, Memphis, TN 38163, USA; 3Laboratory of Muscle Biology and Sarcopenia, Department of Physical Therapy, College of Health Professions, University of Tennessee Health Science Center, Memphis, TN 38163, USA; 4Center for Muscle, Metabolism and Neuropathology, Division of Regenerative and Rehabilitation Sciences, College of Health Professions, University of Tennessee Health Science Center, Memphis, TN 38163, USA; 5The Tennessee Institute of Regenerative Medicine, Memphis, TN 38163, USA

**Keywords:** sarcopenia, skeletal muscle, atrophy, mitochondria, autophagy, mitophagy, aging, mTORC1, dynapenia, caloric restriction, muscle protein synthesis, ubiquitin proteasome, anabolic resistance, rapamycin

## Abstract

Sarcopenia is a debilitating skeletal muscle disease that accelerates in the last decades of life and is characterized by marked deficits in muscle strength, mass, quality, and metabolic health. The multifactorial causes of sarcopenia have proven difficult to treat and involve a complex interplay between environmental factors and intrinsic age-associated changes. It is generally accepted that sarcopenia results in a progressive loss of skeletal muscle function that exceeds the loss of mass, indicating that while loss of muscle mass is important, loss of muscle quality is the primary defect with advanced age. Furthermore, preclinical models have suggested that aged skeletal muscle exhibits defects in cellular quality control such as the degradation of damaged mitochondria. Recent evidence suggests that a dysregulation of proteostasis, an important regulator of cellular quality control, is a significant contributor to the aging-associated declines in muscle quality, function, and mass. Although skeletal muscle mammalian target of rapamycin complex 1 (mTORC1) plays a critical role in cellular control, including skeletal muscle hypertrophy, paradoxically, sustained activation of mTORC1 recapitulates several characteristics of sarcopenia. Pharmaceutical inhibition of mTORC1 as well as caloric restriction significantly improves muscle quality in aged animals, however, the mechanisms controlling cellular proteostasis are not fully known. This information is important for developing effective therapeutic strategies that mitigate or prevent sarcopenia and associated disability. This review identifies recent and historical understanding of the molecular mechanisms of proteostasis driving age-associated muscle loss and suggests potential therapeutic interventions to slow or prevent sarcopenia.

## 1. Introduction

Skeletal muscle comprises over a third of total body mass in young adults [1] and is a primary contributor to whole body metabolic rate and the overwhelming majority of postprandial glucose absorption [2]. Diminished muscle mass is associated with aging, and contributes to greater mortality and metabolic comorbidities [3,4]. Advancements in medical care in the last century have allowed for a remarkable extension of the human lifespan, despite the prevalence of chronic disease and disability in the last decades of life. In recent years, there has been a greater focus on extending the number of years lived in good health and not just increasing the life span per se. This has required an increased emphasis on the importance of muscle mass and quality in maintaining human health.

The concept of “healthy aging” and maintaining independence has been tied to optimizing muscle mass and strength. While muscle mass is critically important to the quality of life, aging is associated with a gradual reduction in muscle mass that plays a contributory role in the development of age-related diseases and disability [5,6]. Sarcopenia is a loss of skeletal muscle mass, quality, and function that is independent of weight loss and usually associated with aging. The European Working Group on Sarcopenia in Older People (EWGSOP) defines sarcopenia as a muscle disease resulting from a combination of low muscle mass and poor muscle function rooted in changes that accrue over a lifetime [7]. Loss of muscle mass has been shown to occur at a rate of 0.7–0.8% per year in the eighth decade of life; however, the loss of strength vastly outpaces the loss of muscle mass [8,9,10].

Sarcopenia is usually, but not always [11,12,13], associated with advanced age. The presence of low muscle mass and function associated with sarcopenia enhances the risk of falls and fractures [14,15], limits mobility [16,17], and increases the risk of disability [18], thus facilitating the necessity for long term assisted living [19]. Sarcopenia also represents a large fiscal burden on global healthcare institutions and government reimbursed healthcare, as patients with sarcopenia can incur larger healthcare cost [20,21]. While the disease sequalae of sarcopenia have been well characterized, the incidence of sarcopenia is expected to become more prominent with the ongoing demographic shift towards an older global population. Indeed, World Health Organization (WHO) projections predict that 1 in 6 people will be over the age of 60 by 2030 [22]. Low muscle mass and function are also central elements of frailty [23,24]. Frailty is an impaired resilience to disease that shares considerable overlap with the clinical manifestations of sarcopenia including an enhanced risk of mortality. Thus, understanding the mechanisms that regulate the loss of muscle mass and function are important if we are to reduce the effects of sarcopenia and frailty.

Well-established contributors to sarcopenia include neuromuscular remodeling [25,26,27], inflammation [28,29,30], apoptotic signaling [31,32,33], and alterations in the hormonal milieu [34,35,36], as well as environmental changes including physical inactivity [27,37,38] and nutritional inadequacies [39,40,41]. Proteostasis and cellular quality control mechanisms have garnered attention as possible points of cellular dysregulation in aging-associated disorders. Protein turnover is critical for tissue integrity, and requires a careful balance between anabolic and catabolic processes. Degradation of misfolded and damaged proteins is necessary to maintain proteome integrity and prevent organelle dysfunction and cellular degeneration. Furthermore, inadequate protein synthesis can result in a failure to replace degraded cellular components and compromised tissue maintenance. Initial lines of evidence supported a role for diminished protein synthesis in response to anabolic stimuli as a major contributor to sarcopenia. Although the mammalian target of rapamycin complex 1 (mTORC1) is critical for protein synthesis and skeletal muscle hypertrophy [42], recent evidence indicates that chronic mTORC1 activity can contribute to deficient cellular quality control and recapitulates several characteristics of sarcopenia including neuromuscular remodeling [43]. Our understanding of how dysregulated protein synthesis and protein degradation in aged skeletal muscle contributes to sarcopenia is incomplete. The present review summarizes previous and recent findings regarding the roles of protein synthesis, degradation, and cellular quality control mechanisms in sarcopenic muscle. Novel and recent studies that describe the interplay between anabolic and catabolic pathways are reviewed. Furthermore, we critically discuss the efficacy of caloric restriction as a treatment for sarcopenia and highlight challenges regarding the putative contributors to proteostasis dysfunction in the development of sarcopenia.

## 2. Skeletal Muscle Anabolism

### 2.1. Anabolic Signaling

Skeletal muscle retains remarkable plasticity by altering cellular size and contractile properties in response to altered stimuli. Muscle fiber size is established as the aggregate result of competition between concomitant rates of anabolic muscle protein synthesis (MPS) and catabolic muscle protein breakdown (MPB). Several anabolic stimuli can enhance MPS, including exercise, dietary amino acids, growth factors, and hormones. Among modulators of MPS, the serine/threonine kinase mammalian target of rapamycin (mTOR) is a critical component of mTORC1 that acts as a central node of regulation (Figure 1). Activation of mTORC1 is tightly regulated by the availability of nutrients and growth factors. The release of insulin by pancreatic β-cells during feeding initiates a cascade of signaling events in skeletal muscle that induces activation of the central effector of the phosphoinositide 3-kinase (PI3K) pathway, Ser/Thr kinase AKT (Akt), also known as protein kinase B (PKB). Insulin-mediated phosphorylation of Akt facilitates activation of mTORC1 through phosphorylation and inactivation of the tuberous sclerosis complex 1/2 (TSC1/2) [44], which acts as a GTPase-activating protein (GAP) for Ras homolog enriched in brain (Rheb) at the lysosomal surface and thereby prevents mTORC1 activation (Figure 1). Upon insulin-mediated phosphorylation by Akt, TSC1/2 dissociates from the lysosome and allows for Rheb activation of mTORC1 [45].

In addition to prompting the release of insulin, feeding also replenishes intracellular nutrients that can induce mTORC1 activation, in particular amino acids. Cytosolic availability of leucine is a potent signal for mTORC1 activation in large part through the leucine sensor Sestrin2. The availability of leucine reduces Sestrin2 activity, which relieves inhibition of the protein complex GTPase-activating protein activity toward Rags 2 (GATOR2) [46,47]. Uninhibited GATOR2 antagonizes the action of GTPase-activating protein activity toward Rags-1 (GATOR1), thereby relieving GATOR1 GAP activity on Rag guanosine triphosphatases [46] and allowing for their recruitment of mTORC1 to the lysosome in close proximity to Rheb. Activation of mTORC1 is also highly sensitive to arginine, which similarly to leucine, can relieve inhibition of GATOR2 through a cytosolic arginine sensor for mTORC1 subunit 1 (CASTOR1) dependent mechanism [48,49] (Figure 1). As such, a lysosome localized coincidence detector regulates cellular protein synthesis via mTORC1 by integrating extracellular growth signals with the intracellular nutrient status. Activation of mTORC1 inhibits degradative processes such as autophagy and stimulates mRNA translation in part through phosphorylation of targets eukaryotic translation initiation factor 4E (eIF4E)-binding protein 1 (4E-BP1), Eukaryotic elongation factor 2 (eEF2) kinase, and p70 ribosomal protein S6 kinase (p70^S6K1^). Phosphorylation of 4E-BP1 by mTORC1 results in the disassociation of 4E-BP1 from eIF4E and allows for the formation of the translation initiation complex at the 5′ end of mRNA to facilitate recruitment of the ribosomal machinery and cap dependent mRNA translation [50]. mTORC1 also increases protein synthesis through its enhancement of translation elongation via inhibitory phosphorylation of eEF2 kinase [51]. Activation of p70^S6K^ by mTORC1 also boosts protein synthesis through augmentations in ribosomal biogenesis [52], translation initiation [53], and elongation [54].

### 2.2. Skeletal Muscle Anabolic Resistance

#### 2.2.1. Amino Acid Induced MPS

Muscle atrophy occurs when myocytes have reduced protein accretion that results in reduced cellular size as the aggregate result of competition between reduced concomitant rates of MPS and increased MPB. Anabolic resistance refers to a blunted capacity to enhance MPS in response to an anabolic stimulus. Associations between aging and anabolic resistance have been under investigation for decades and have indicated that a perturbed skeletal muscle anabolic response to a nutritional stimulus exists in aged individuals [55,56]. While basal skeletal muscle fractional synthetic rates have been shown to be similar between young and aged individuals [56,57,58,59], the sensitivity and responsiveness of MPS have been reported to be reduced in the elderly in response to hyperaminoacidemia [56,57,60]. This suggests that senescent muscle may require a relatively high concentration of amino acids [61] to mount an adequate anabolic response. While the precise mechanism(s) that regulate the blunting of MPS in response to feeding in aging is/are not clear, a likely contributor is the reduction in physical activity that typically occurs with aging [62]. It is known that reduced physical activity can diminish the protein synthetic response to feeding [63] and this may play a large role the development of sarcopenia.

While there is significant support for an aging-suppression of MPS, not all studies have observed a blunted maximal MPS response to high doses of amino acids with elevated ages [64,65]. Furthermore, the concept that aging induces a reduced anabolic response to amino acid intake has also recently been challenged [66]. Nevertheless, while not a universal finding, the majority of studies support a role for attenuated MPS as a potential contributor to sarcopenia in aging.

#### 2.2.2. MPS and Sarcopenic Obesity

The co-occurrence of age-related muscle loss and obesity, termed sarcopenic obesity, is a growing health concern. Sarcopenic obesity enhances the risk of developing type 2 diabetes mellitus and insulin resistance [37,67]. The elderly exhibit greater intramuscular lipid deposition than younger body mass and fat mass matched control subjects [68]. Aged animals also exhibit exacerbated insulin resistance when fed an obesogenic diet, which is concomitant with greater susceptibility to the accumulation of intramyocellular lipids [69,70]. Importantly, both lipid infusions and obesity have been shown to reduce the MPS response to amino acid and protein stimulation [71,72,73]. Investigations regarding the impact of coincident obesity and aging on the skeletal muscle protein synthetic response to protein ingestion have revealed that myofibrillar fractional synthetic rate is enhanced in young lean subjects and, to a significantly lesser degree, old lean subjects. However, old obese subjects fail to enhance post prandial myofibrillar fractional synthetic rate, indicating that obesity further dampens aging associated anabolic resistance to feeding [74]. Interestingly, while it is generally agreed upon that muscle loss with aging occurs at the expense of fast twitch type II fibers, it is not well understood if post prandial protein synthesis exhibits a fiber type specific deficit in aged muscle or how obesity may modulate this relationship. However, evidence exists to suggest that age-associated muscle atrophy may not be as restricted solely to fast twitch fibers, as was once thought to be the case in sarcopenia [75].

#### 2.2.3. Gastrointestinal–Muscle Axis

Recent observations have linked the gut-muscle axis as a potential contributor and therapeutic target for aging associated anabolic resistance. While still in its infancy, several studies in rodents have linked alterations to the gut microbiota to aging and sarcopenia [76,77]. Indeed, the gut microbiome appears to have a major impact on skeletal muscle. For example, germ-free mice that are microbiologically sterile exhibit reduced skeletal muscle mass alongside diminished gene expression for oxidative metabolism. Importantly, these deficits are partially reversed by microbiota transplant from pathogen free mice [78]. Furthermore, gnotobiotic mice, which are germ-free mice transplanted with a known microbe, have demonstrated a causal link between specific microbes and skeletal muscle function. Gnotobiotic mice exhibited enhanced physical function both in the untrained state and after 4 weeks of exercise training when compared to germ-free mice, indicating an important role for gut microorganisms in physical function and exercise adaptations [79]. While mechanistic investigations regarding the interaction between gut microbes and skeletal muscle anabolism are sparse, analysis of the gastrointestinal microbiome of chickens revealed significant differences in microbe composition between chickens with low- and high-feed conversion ratios [80], a measure of the ratio of feed consumed to weight gained. These data imply that the gut microbiome composition can modulate the nutritive value of food consumed and the degree of anabolic growth.

#### 2.2.4. Microbiome and Inflammation in Sarcopenia

It is interesting to note that a greater proportion of germ-free mice survived to old age than pathogen-free mice that are housed under standard laboratory conditions, which present with a more pronounced inflammatory state and enhanced gut permeability with increased age [81]. Germ free mice co-housed with aged pathogen-free mice exhibited a greater degree of intestinal permeability and circulating tumor necrosis factor α (TNFα) than mice that were co-housed with young pathogen free mice. This suggests that microbiome-associated changes with aging can influence the systemic inflammation of the host animal. Aging is also associated with reduced colonic mucosal barrier thickness [82] and a greater circulating concentrations of lipopolysaccharide [83]. Thus, it is possible that changes in the microbiome or gut health may be involved in heightened systemic inflammation associated with aging. The relationship between sarcopenia, aging, and chronic low-grade inflammation is well established and multiple lines of evidence have shown that chronic inflammation is deleterious to skeletal muscle health and increases protein turnover. Thus, alterations in the microbiome or gut function provide an attractive and plausible link to aging, inflammation, and skeletal muscle health.

It is important to note that while skeletal muscle is modulated by the microbiome, the gut microbiome can be modified by diet, antibiotics, and dietary supplements. The human microbiota has been reported to be similar between young and aged humans but there are larger differences between the young and the very old [84]. Furthermore, a study of long-term care and community dwelling seniors found that the diversity of the gut microbiome was significantly lower in individuals under long-term care when compared to similarly aged community dwelling seniors [85], highlighting the importance of considering environmental factors such as diet that may differ between individuals when investigating age-associated dysbiosis. These studies suggest that diet and age can affect skeletal muscle health both directly through the dietary supply of nutrients and indirectly through the dietary effect on the microbiome at any age. It is possible that alterations to the gut microbiome may also partly be due to environmental factors associated with frailty and sarcopenia in adults that may exhibit physical inactivity, alterations in diet, and consume medications for existing comorbidities.

### 2.3. Aging and Skeletal Muscle Recovery from Disuse

Skeletal muscle necessitates contractile and neural stimuli to maintain tissue integrity [86,87]. Extended periods of disuse such as bed rest, limb immobilization, and spaceflight can rapidly produce deficits in muscle mass. Muscle disuse induces a loss of contractile function leading to muscle atrophy, mitochondrial impairments, and functional decline [88,89]. The elderly are prone to sequential periods of disuse, due to more frequent falls and hospitalization than younger healthy subjects. In addition, the elderly exhibit poor regeneration of muscle tissue after an injury or disuse as compared to young subjects or animal models [90,91,92], which can potentially hasten the trajectory towards sarcopenia after repeated bouts of bed rest or immobilization. It is generally accepted that skeletal muscle atrophy during disuse occurs due to reductions in MPS and a heightened MPB [93], thereby compromising tissue maintenance. Importantly, aged animals show marked atrophy after disuse [94,95,96,97,98] and delayed restoration of contractile tissue [94,99,100,101], mitochondrial function, and dysfunctional β-oxidation after disuse compared to young animals [102]. Temporal assessments of the genetic response to hypertrophic stimuli have revealed metabolic gene expression to be perturbed by aging [103], which may play into the impaired response to reloading. Peroxisome proliferator-activated receptor-gamma coactivator (PGC)-1α/β-knockout mice have delayed force output recovery after disuse, indicating a crucial role for mitochondria in the functional recovery of skeletal muscle [104]. While protein ubiquitination and the expression of atrogenes muscle-specific RING finger protein 1 (MuRF1) and Atrogin1 are enhanced in aged animals during recovery after hindlimb suspension, proteosome activity and autophagic flux in some muscles are only enhanced in young animals in the reloading phase [105,106]. The accumulation of ubiquitinated proteins and reduced proteasomal activity in combination with an elevation of endoplasmic reticulum (ER) stress markers suggests dysregulated proteostasis in old animals [105], which could feasibly contribute to inadequate regeneration after disuse in geriatric muscle. Both protein synthesis and protein degradation are ATP-dependent processes. Thus, skeletal muscle remodeling is dependent upon adequate mitochondria number and function to supply ATP for protein synthesis and degradation. It has been estimated that the energetic cost of protein synthesis amounts to about 20% of basal ATP produced by oxidative phosphorylation [107,108]. While humans typically have enough fuel stores to sustain metabolic processes for several days to weeks, energy production is constrained by the rate of mitochondrial bioenergetics. While disuse impairs mitochondrial function [102,109], mitochondrial adaptations to exercise and activity increase mitochondrial number and function [110] and provide an optimal environment for muscle repair. Furthermore, muscle reloading in young animals reverses disuse-induced mitochondrial dysfunction [102,109]. In addition, both voluntary wheel running and a cocktail of mitochondrial targeted nutrients improve muscle regeneration after disuse [109,111], suggesting the restoration of mitochondrial function may facilitate tissue regrowth.

Regulators of mitochondrial volume and function are suppressed during muscle disuse as energy demands and contractile activity diminish. However, it is unclear if mitochondrial impairments during recovery may contribute to age-associated regenerative defects in skeletal muscle and contribute to sarcopenia. It is important to note that some studies have found that the amount of atrophy that occurs during disuse is not exacerbated by age [106,112] and the absolute extent of atrophy during disuse is sometimes more pronounced in young muscle [113], although this may not always be the case during short term periods of disuse [114]. This may be due in part because muscles in aged adult animals or humans are sarcopenic and have a smaller muscle mass and therefore have less protein to lose during disuse even if the rate of muscle loss is the same in young and old animals.

Another possibility is that the rate of MPS after disuse may be altered with aging. This is supported by observations that the larger muscles in young animals exhibit a more pronounced drop in MPS during disuse when compared to smaller muscles in aged animals [106]. When MPS was measured during recovery from disuse in aged animals, it was found that the rate of MPS was not diminished in 28-month-old when compared to 24-month-old rats, and in some muscles was significantly elevated with older age [106]. Furthermore, MPS was found to be elevated in old but not young rats during reloading despite blunted recovery [105]. If MPS is not impaired with aging, yet protein accretion is impaired in older hosts, then it is possible that geriatric muscle has a reduced ability to coordinate processes of protein synthesis, protein folding, and degradation of contractile tissue during reloading. A dysregulation of protein assembly in aging would ultimately compromise muscle remodeling and the reacquisition of muscle mass and this could contribute to sarcopenia if muscle mass and function are not restored after disuse.

### 2.4. Sarcopenia and mTORC1 Signaling

Although mTORC1 is an important regulator of muscle growth, recent evidence has implicated that sustained mTORC1 activation may be a potential contributor to age-associated myopathy (Figure 2). Despite reported insensitivity of MPS to external stimuli in aging, there are several reports suggesting that geriatric skeletal muscle exhibits signs of greater basal mTORC1 activity [43,115,116,117,118,119].

How alterations in senescent muscle drive greater basal mTORC1 activity or how sustained mTORC1 activity may differ between young and aged muscle have yet to be determined. Nevertheless, several studies have provided some clues to how mTORC1 could be elevated in aged muscle. For example, data from a transgenic mouse model that allows for the muscle-specific inducible expression of activated Akt has shown that activation of Akt (and presumably also mTORC1) for three weeks, results in greater muscle mass in young [120] and old (24 months) mice [115]. However, in contrast to young mice, old mice exhibited a reduction in force production when normalized to muscle mass (lower specific force), which was concomitant with morphological signs of fiber degeneration and immune cell infiltration [115]. These data imply that the chronic activation of the same anabolic signaling pathways (i.e., mTORC1) can produce very different effects in aged and young muscle. In a separate study, acute activation of mTORC1 in skeletal muscle by shRNA knockdown of TSC produced fiber hypertrophy [121], while sustained mTORC1 activation by genetic skeletal muscle TSC1-knockout, diminished muscle size and reduced total lean mass [121,122,123]. Indeed, inactivation of mTORC1 by muscle specific regulatory associated protein of mTOR (RAPTOR)-knockout also produces skeletal muscle atrophy [124]. Taken together, it appears that the proper balance of mTORC1 is required for optimal muscle health because both chronic loss and hyperactivation of mTORC1 can be deleterious for maintenance of skeletal muscle mass. The proposed model of mTORC1 dysregulation in aging leading to muscle atrophy is shown in Figure 2.

There is a complex regulation of muscle mass in aging because inhibition of chronically active mTORC1 with either rapamycin or low dose rapalogs attenuates aging-associated loss of muscle function and muscle mass in some, but not all, muscles [43,117,125]. Mechanistically, it is possible that hyperactivity of mTORC1 may negatively impact neuromuscular junction function, as both skeletal muscle specific TSC1-knockout mice and aged mice exhibit morphological alterations that are characteristic of neuromuscular junction instability. Importantly, neuromuscular deficits in mTORC1 hyperactive muscle and in aged mice are both partially reversed by mTORC1 inhibition via rapamycin administration [43]. Dysfunction of the motor neuron is a known contributor to the development of sarcopenia [126]. These findings support the idea that sustained mTORC1 activation in geriatric muscle can contribute to neuromuscular junction instability (Figure 2) and subsequently muscle atrophy and functional deficits, which may be partially reversed by rapamycin treatment. While rapamycin has been shown to produce favorable effects on sarcopenic muscle, it remains unclear if similarly inhibiting mTORC1 during reloading after disuse would enhance recovery and ameliorate ER stress in aged muscle (Figure 2).

## 3. Skeletal Muscle Catabolism

### 3.1. The Ubiquitin Proteasome System

#### 3.1.1. Ubiquitin Proteasome Degradation of Cellular Proteins

Protein turnover is critical for cellular quality control under both basal and physiologically stressful conditions to maintain the integrity of the proteome. Degradation of cellular proteins occurs primarily through the ubiquitin proteasome system (UPS), autophagy, and to a lesser extent the Ca^2+^ dependent calpain system and caspase-3 [127,128]. Inadequate protein and organellar degradation can result in the accumulation of misfolded protein aggregates and dysfunctional cellular organelles. The primary component of the UPS is the 26S proteasome which is comprised of a core proteolytic 20S complex and the regulatory 19S complex capping one end of the 20S, although both ends of the core 20S can be capped to form the 30S proteasome [129]. The 19S regulatory particle is involved in the process of substrate engagement and subsequent ATP-dependent processive translocation and unfolding of the target polypeptide into the degradative 20S barrel subunit [130]. Proteins marked for degradation are recognized by the 26S proteasome through the presence of conjugated polyubiquitin tails.

Ubiquitination of damaged or misfolded proteins is a complex and highly regulated process involving a molecular hierarchy of enzymes involved in ubiquitin activation, conjugation, and ligation (reviewed in [130,131]). Briefly, conjugation of ubiquitin is first initiated by the ubiquitin-activating enzyme (E1), an ATP-dependent process that that results in a high energy thioester bond between E1 and ubiquitin. Secondly, there is the transfer of the ubiquitin-thioester to a ubiquitin conjugating protein (E2), which is followed by the conjugation of ubiquitin by E3 ubiquitin ligases to target proteins. While protein ubiquitination also serves other important non-proteasomal degradation roles such as, alterations of protein localization and function, these roles will not be covered in this review.

Relative to the small number of proteins within the human E1 (*n* = 2) and E2 (*n* = ~40) family, there are hundreds of different E3 ubiquitin ligases that confer specificity to the UPS, allowing for tight control of substrate ubiquitination [131]. Activation of the UPS has been implicated as a contributor to skeletal muscle atrophy under several pathological conditions including aging-induced sarcopenia [132,133], type 1 diabetes [134], muscle disuse [135,136,137,138,139], and fasting [138,140]. The two best-characterized E3 ubiquitin ligases implicated in skeletal muscle atrophy are Atrogin1 and MuRF1 [137]. Atrogin1 has been shown to mediate degradation of myogenic factors Myogenic Differentiation 1 (MyoD1) [141,142], myogenin [143], and eukaryotic translation initiation factor 3 subunit F (eIF3-f) [144]. Additionally, both Atrogin1 and MuRF1 have been shown to interact with sarcomeric proteins [145,146,147]. Furthermore, MuRF1 has also been implicated in the regulation of acetylcholine receptors at the neuromuscular junction [148]. Expression of atrogenes Atrogin1 and MuRF1 is regulated by Forkhead box O (FoxO) transcription factor translocation into the nucleus [138,149,150], although FoxO has also been shown to regulate autophagy [151]. Importantly, Akt can prevent FoxO nuclear localization [150], while FoxO activity is enhanced by energy deprivation through AMP-activated protein kinase (AMPK) [152,153].

#### 3.1.2. The Ubiquitin Proteasome System in Sarcopenia

Given the ubiquitous FoxO-mediated expression of Atrogin1 and MuRF1 in a range of atrophic conditions, the expression of Atrogin1 and MuRF1 has often been used as a proxy measure for protein breakdown in skeletal muscle. However, it is important to note that enhanced expression of Atrogin1 and MuRF1, and even enhanced protein ubiquitination, is not synonymous with enhanced proteasome activity. Indeed, muscles from aged rats subjected to hindlimb unloading followed by reloading exhibit greater Atrogin1 and MuRF1 gene expression alongside elevated protein ubiquitination as compared to young adult animals; however, proteasome activity was generally unchanged or reduced throughout unloading and reloading in the older rodents [105]. These responses differed from muscles in young adult animals and the responses can even differ between muscles of older animals. This may represent a mismatch between ubiquitination and proteosome activity in geriatric muscle and potentially impaired UPS function during skeletal muscle reloading with aging [132,133].

The UPS system is tightly regulated to ensure proper protein turnover which is necessary to maintain a high quality of the muscle protein. Loss of protein quality control mechanisms is a hallmark of aging and a decline in protein turnover and the function of the UPS has long been thought to contribute to age-related cellular dysfunction [154]. The ability to rapidly degrade proteins in molecular signaling pathways is critical to avoid aberrant signaling and maintain proteome stability. Indeed, while overactivation of the UPS results in skeletal muscle atrophy, inadequate proteasome function in the skeletal muscle of Rpt3-knockout mice severely retards growth, resulting in reduced muscle mass, function, and signs of fiber degeneration concomitant with enhanced expression of Atrogin1, MuRF1, and protein ubiquitination [155]. The elevation in atrogene expression and protein ubiquitination may indicate a compensatory effort to ramp up protein degradation in the absence of adequate proteostasis. Similarly, MuRF1 and MuRF3 double-knockout mice exhibit myofiber protein aggregates, ultrastructural abnormalities, Z-line streaming, and pronounced functional deficits despite greater muscle mass [156]. Somewhat counterintuitively, MuRF1 null mice that are significantly protected from denervation induced atrophy also display a greater enhancement of skeletal muscle proteasome activity than wild type mice [157]. Similarly, while wildtype mice present a reduction in proteasome activity by 24 months of age, MuRF1 null mice exhibit enhanced proteasome activity and preserved skeletal muscle mass and fiber cross sectional area. Aged wildtype mice also showed signs of enhanced ER stress with age, which was attenuated in MuRF1 null mice [158]. Interestingly, despite a preservation of muscle mass, old MuRF1 null mice presented functional deficits in nerve-stimulated muscle force production when compared to age-matched wildtype mice. Given the role of MuRF1 in acetylcholine receptor regulation [148], it is possible that the observed defect may originate from inadequate neuromuscular remodeling.

Whether skeletal muscle UPS activity is enhanced or diminished with aging is controversial, with several studies pointing towards an increase [159,160,161], whereas others report a decrease of UPS activity in geriatric muscles [158,162,163]. Despite this lack of consensus, recent evidence has provided a surprising connection between elevated skeletal muscle mTORC1 signaling and activation of degradation pathways including the UPS system. Kaiser et al. [164] observed that mTORC1 hyperactivation in skeletal muscle of TSC1 null mice (a mouse model that recapitulates several characteristics of an aged phenotype), is concomitant with marked increases in several components of the UPS. Skeletal muscle from TSC1-knockout mice exhibit greater expression of several atrogenes and components of the 26S proteasome that is reversed by acute 3-day treatment with rapamycin. Remarkably, transcriptomic changes observed in muscles from TSC1-knockout mice after rapamycin treatment were highly similar to those observed in 30-month-old rapamycin treated mice [43,164]. Hyperactivation of mTORC1 in skeletal muscle of TSC1 muscle-knockout mice was concomitant with enhanced protein abundance of Nuclear Factor Erythroid 2-Like 1 (NRF1); a potent positive regulator of proteasomal gene expression. Knocking down NRF1 expression via shRNA reduced proteasomal protein abundance and catalytic activity, however this was insufficient to rescue muscle mass in TSC1 skeletal muscle-knockout mice [164].

Importantly, greater mTORC1 activity can impede AKT activation through p70^S6K1^-mediated phosphorylation of Insulin receptor substrate 1 (IRS-1) [165], thereby inhibiting downstream AKT activation. To test whether inhibition of AKT by hyperactive mTORC1 in TSC1-knockout mice was responsible for the apparent muscle atrophy, TSC1 muscle-knockout mice were bred with mutant AKT-TG mice that express a tamoxifen inducible form of active AKT. In line with AKT’s role as a negative regulator of FoxO and atrogene expression, AKT activation induced muscle growth in TSC1-knockout muscle. These data indicate that AKT-mediated suppression of FoxO targets, but not suppression of NRF1 and proteasomal gene expression, rescues muscle atrophy in the background of mTORC1 hyperactivation. As aging mice have been reported to have chronically high levels of mTORC1, as discussed above, the reversal of the suppression of AKT-regulated UPS by mTORC1 may provide a potential target for therapeutic reduction of sarcopenia in aging. However, simply increasing AKT may not be an adequate approach for treating sarcopenia because despite enhanced muscle size, sustained hyperactivation of AKT was reported to produce vacuolated fibers that are characteristic of late onset myopathy during mTORC1 hyperactivation. Nevertheless, the elegant work of Kaiser et al. [164] showed that proteasomal upregulation is an attempt to maintain proteostasis during mTORC1 hyperactivation (and diminished autophagy) and inhibition of FoxO-mediated gene expression results in early onset myopathy.

### 3.2. Autophagy and Mitophagy

#### 3.2.1. Autophagy and Mitophagy Signaling

Similar to the UPS, autophagic degradation represents another major arm of protein quality control, to ensure cellular clearance of defective proteins and organelles as well as to recycle substrates during nutrient scarcity. A basal level of autophagy is important for cellular homeostasis, and the autophagic machinery is responsive to several stimuli including fasting [166,167] and exercise [168]. However, autophagy is elevated in response to oxidative stress [169], ER stress [170], and to pathological states such as cancer cachexia [171]. There exist three well-characterized types of autophagy which include microautophagy, chaperone-mediated autophagy, and macroautophagy. Macroautophagy primarily manages the clearance of cellular organelles and protein aggregates via engulfment by the autophagosome and degradation in lytic compartments through fusion with the lysosome. For the purposes of this review, macroautophagy will be referred to as autophagy (Figure 3).

The initiation of autophagy and autophagosome biogenesis is largely regulated by unc-51 like autophagy activating kinase 1 (ULK1) and to a lesser extent its paralog unc-51 like autophagy activating kinase 2 (ULK2). ULK1 is a serine/threonine kinase that is the core component of the initiation complex alongside autophagy related (ATG)13, Focal adhesion kinase family-interacting protein of 200 kDa (FIP200), and ATG101 [172,173,174,175]. During nutrient deprivation, autophagy is rapidly stimulated to allow for bulk degradation to supply provisions of amino acids for continued synthesis. Autophagy is negatively regulated by mTORC1 during nutritional and energetic surplus (Figure 3). However, amino acid insufficiency relieves mTORC1-mediated inhibitory phosphorylation of ULK1 and ATG13 [175], enhancing autophagy. Importantly, cellular energy status is also conveyed to the initiation complex through the energy sensor AMPK, which can inhibit mTORC1 or directly activate ULK1 via phosphorylation [176]. Apart from direct inhibitory phosphorylation of ULK1, mTORC1 also regulates ULK1 stability through inhibition of autophagy/Beclin 1 regulator (AMBRA1). AMBRA1 acts to facilitate TNF receptor associated factor 6 (TRAF6)-mediated ubiquitination of ULK1 and subsequent protein stabilization [177]. Upon ULK1 activation, ULK1 phosphorylates AMBRA1 and allows for the localization of the BECLIN 1-VPS34 core complex to autophagosome initiation sites primarily at the ER [178]. In addition to inducing the translocation of the BECLIN 1-VPS34 core complex, ULK1 also activates BECLIN 1 (Figure 3) and stimulates VPS34 class III phosphatidylinositol 3-kinase activity [179]. Production of phosphatidylinositol 3-phosphate at the site of phagophore initiation is a critical step that engages the WD-repeat protein interacting with phosphoinositides (WIPI) family of proteins which serve as effectors that recruit several ATG proteins and subsequently LC3 lipidation [180]. Lipidation of LC3 via conjugation to phosphatidylethanolamine plays a critical role in autophagosome formation through expansion of the isolation membrane [126,181,182,183].

While excessive autophagy can contribute to atrophy [151], in recent years selective autophagy of mitochondria, termed mitophagy, has gained attention as a necessary and critical quality control step for skeletal muscle health (Figure 3). Mitochondrial dysfunction that results in the loss of the mitochondrial membrane potential (Δψm) triggers a cascade of events that culminate in sequestration and autophagic clearance of damaged mitochondria. Under normal conditions, the serine/threonine kinase PTEN induced kinase 1 (PINK1) is sequestered within the mitochondrion [184] and its degradation depends on intact Δψm [185]. Upon mitochondrial damage and membrane depolarization, PINK1 is stabilized and accumulates in the outer mitochondrial membrane (OMM) [184,185,186] where it recruits the ubiquitin ligase Parkin (Figure 3). Parkin is normally cytosolic and inactive under control conditions [185]. Upon mitochondrial localization, Parkin is activated and decorates OMM proteins with polyubiquitin chains [187] (Figure 3), thereby promoting mitophagy through interactions with autophagy adapter proteins [188,189].

Investigations regarding whether skeletal muscle autophagy is reduced or enhanced with age have reached mixed conclusions [190,191,192,193,194,195]. While the consensus is that autophagy is blunted with aging, evidence for altered autophagy has primarily relied on the quantification of basal protein abundance of several autophagy effectors. However, changes to the abundance of autophagy regulators such as LC3-I/LC3-II do not accurately reflect autophagic flux within the tissue and instead represent a fixed snapshot in time of a highly dynamic process. Indeed, the LC3-I/LC3-II content may be enhanced due to greater LC3 lipidation or diminished lysosomal degradation. Autophagy can be more accurately estimated using autophagy inhibitors such as colchicine, allowing for measurements of accumulated autophagy effectors as an output for autophagy flux [196].

#### 3.2.2. Defective Autophagy Impairs Skeletal Muscle Function and Mass

Measurements of autophagy flux have shown a trend for enhanced basal autophagy in aged muscle [105,193,197]. Nonetheless, it is intriguing that two effective interventions for mitigation of sarcopenia and aging, namely caloric restriction [198] and exercise [168,195,199,200], are known to stimulate skeletal muscle autophagy. Additionally, reducing autophagy in skeletal muscle recapitulates several aspects of sarcopenia. For example, autophagy deficient skeletal muscle of ATG7-knockout mice exhibit signs of myopathy alongside compromised skeletal muscle mass and function [201]. Skeletal muscle specific knockout of ATG7 in aged mice similarly reduces muscle mass and produces signs of neuromuscular junction dysfunction [195].

Skeletal muscle AMPK-knockout mice also have defective autophagy and exhibit age-related muscle dysfunction associated with signs of fiber degeneration, diminished force production, and greater mitochondrial size and Parkin accumulation at 18 months of age [202]. These data imply that relative to age matched wild types, reduced autophagy in skeletal muscle of AMPK-knockout mice results in impaired muscle function and mitochondrial clearance, which manifests as myopathy in an age-dependent manner [202]. Additionally, evidence from the literature indicates that enhancing autophagy ameliorates the aging phenotype. While activation of AMPK is known to inhibit mTORC1 and protein synthesis, it is interesting to note that muscle atrophy in obese sarcopenic rats was ameliorated by resveratrol, which was also found to prevent myotube atrophy *in vitro* partially via the PKA/LKB1/AMPK pathway [203].

Skeletal muscle expression of FoxO in drosophila enhances autophagy and delays the aging associated accumulation of protein aggregates in skeletal muscle [204]. In contrast, skeletal muscle FoxO1/3/4 triple-knockout mice exhibit spared muscle mass and preserved specific force, a functional measure of muscle quality, during aging without apparent signs of muscle degeneration, despite dramatically greater fiber central nucleation [205]. While autophagy flux was not compared in aged control and triple-knockout mice, young triple-knockout mice did not present an apparent defect in muscle autophagy. Thus, the preservation of skeletal muscle mass and function in FoxO triple-knockout mice may be due to an observed conservation of mitochondrial function without apparent changes to the autophagy machinery.

#### 3.2.3. mTORC1 and Autophagy

One potential mechanism by which autophagy may be impaired in aged skeletal muscle is via sustained activation of mTORC1. Comparisons of autophagy and mTORC1 signaling in skeletal muscle of mice that are freely fed revealed little difference between young and old animals. However, upon fasting, skeletal muscle from young mice displayed diminished phosphorylation of AKT and the mTORC1 targets p70^S6K1^, ribosomal protein S6 (RPS6), and 4E-BP1. Notably, fasting-induced changes were absent in aged mice. Furthermore, mTORC1 phosphorylation of ULK1 (Ser757) was reduced by fasting in young but not aged mice [116]. Mirroring the sustained mTORC1 activity and reduced autophagic signaling that has been reported in aged animals, skeletal muscle TSC1-knockout animals exhibit impaired autophagy that is concomitant with late onset myopathy, both of which are reversed by rapamycin treatment [206]. Indeed, the enhanced UPS activity present in skeletal muscle TSC1-knockout mice may be an attempt to compensate for impaired autophagic clearance [180]. A recent study by Crombie et al. [207] revealed a previously unknown aspect of mTORC1 regulation of autophagy in skeletal muscle. Interestingly, activation of 4E-BP1 but not inhibition of p70^S6K1^ ameliorates the sarcopenic phenotype of skeletal muscle TSC1-knockout mice [207]. Despite reduced MPS with both p70^S6K1^ inhibition and 4E-BP1 activation, only activation of 4E-BP1 in TSC1-knockout muscle improved muscle size, function, and the induction of autophagy by fasting despite no change in ULK1 phosphorylation. TSC1 muscle-knockout mice exhibited greater protein aggregates and histological indications of accumulated mitochondria and lipofuscin; characteristic of blunted mitophagy and lysosome function. Importantly, activation of 4E-BP1 ameliorated signs of mitochondrial and lysosomal dysfunction present with mTORC1 hyperactivation, suggesting a critical role for 4E-BP1 modulation of autophagy that is independent of mTORC1-mediated ULK1 phosphorylation.

#### 3.2.4. Autophagy and Mitochondrial Dysfunction

It is well established that mitochondrial dysfunction is a core characteristic of sarcopenia [126]. Whether aged skeletal muscle mitochondria exhibit a bioenergetic defect is debated, as respiration measured in isolated mitochondria produce a pronounced age associated defect that is less apparent in *in situ* preparations that maintain mitochondria in their native cellular environment [208]. This apparent discrepancy may be due to greater fragility of aged mitochondria that impairs their resilience to standard isolation procedures, thereby exacerbating respirometric defects [208,209]. Nonetheless, a greater susceptibility to apoptosis is a well-documented characteristic of mitochondria from aged muscle [126,210] which may be secondary to impaired mitophagy [210,211]. Additionally, while mitochondrial respiration is contingent on physical activity status in aged adults, sensitivity of mitochondria to permeability transition is enhanced by old age regardless of activity status [27], indicating that perturbations to mitochondrial health and apoptotic susceptibility exists in aged skeletal muscle independent of intrinsic bioenergetic capacity. While mitochondrial bioenergetic defects in aged skeletal muscle are not a universal finding, it is important to recognize that high-resolution respirometry, often used to measure mitochondrial function, commonly employs substrate-uncoupler-inhibitor-titration (SUIT) protocols that measure mitochondrial respiration in the presence of kinetically saturating concentrations of ADP that are several-fold higher than physiological ADP concentrations. Indeed, when tested at physiologically relevant ADP concentrations, *in situ* preparations of human aged skeletal muscle exhibit a respirometric defect and a higher Km for ADP when compared to young muscle [212], suggesting impaired ADP sensitivity in mitochondria from aged muscle. High-fat-diet feeding, which results in obesity and impairs mitochondrial health [213], also induces skeletal muscle respirometric defects at submaximal but not maximal ADP concentrations in rodents, despite the greater abundance of mitochondrial proteins [214]. Interestingly, in a murine model of sarcopenic obesity, mitochondrial uncoupling enhanced markers of mitophagy and improved skeletal muscle mass and function [215]. These data imply that the removal of dysfunctional mitochondria is critical for improving muscle function during sarcopenic obesity. Given that mitochondria exhibit a high spare capacity for ATP production during repeated muscle contraction, it is unclear if marginal reductions in bioenergetic function would perturb cellular homeostasis or produce an energetic defect at rest. However, the potential for greater reactive oxygen species production can induce damage to mitochondrial components for which there may be insufficient mitophagy to mitigate in aged muscle.

#### 3.2.5. Ca^2+^ Dysregulation in Sarcopenia

Oxidative stress produced by the accumulation of dysfunctional mitochondria can also damage cellular structures that are near the damaged mitochondria. Indeed, in muscle, mitochondria share an intimate role with the ER/sarcoplasmic reticulum (SR) and influence Ca^2+^ homeostasis. Skeletal muscle function relies heavily on the propagation of signals from the neuromuscular junction to the transverse tubules to the SR, a specialized portion of the ER in striated muscle that mediates the release of Ca^2+^ ions through the ryanodine receptor to initiate muscle contraction. Disruption of Ca^2+^ handling by the SR can lead to elevated cytosolic Ca^2+^ and in turn, exacerbates mitochondrial reactive oxygen species (ROS) production. Oxidation of the ryanodine receptor is greater in aged than in young muscle, which can result in Ca^2+^ “leakage” from the SR into the cytoplasm [216]. Muscles from mice expressing a mutated form of the ryanodine receptor that enhances SR Ca^2+^ leakage present similar contractile deficits as aged mice [216]. In agreement with these findings, Delrio-Lorenzo et al. [217] found that skeletal muscle SR Ca^2+^ concentration decreases with age in drosophila melanogaster. Furthermore, the decrease of Ca^2+^ concentration was closely correlated with reduced physical function and not observed in neurons, highlighting a tissue specific dysregulation of Ca^2+^ homeostasis [217]. In addition to dysfunction at the ryanodine receptor, sarcopenia is also associated with reduced sarcoplasmic reticulum Ca^2+^ ATPase (SERCA) pump activity [218]. The SERCA pump is important for reuptake of Ca^2+^ from the cytosol to the SR at the end of an electrically evoked stimulus. Interestingly, treatment of aged mice with the allosteric SERCA activator CDN1163 improved indices of muscle function and attenuated mitochondrial dysfunction [218]. Furthermore, CuZnSOD deficient (Sod1^-/-^) mice, a model of premature aging, also exhibited reduced SERCA pump activity and skeletal muscle contractile deficits that are rescued by CDN1163 treatment [219]. Taken together, it appears that oxidative stress during advanced age perturbs SERCA pump function and elevates cytosolic skeletal muscle Ca^2+^, which can amplify mitochondrial dysfunction and contribute to declines in contractile function.

In addition to dysfunctional Ca^2+^ reuptake, there is evidence that store-operated Ca^2+^ entry (SOCE) is inefficient in sarcopenic muscle and may contribute to declines in skeletal muscle performance. SOCE is critical to maintain intracellular Ca^2+^ homeostasis as ER/SR Ca^2+^ stores are depleted. In brief, reductions in ER/SR Ca^2+^ are sensed by the Ca^2+^ sensor stromal interaction molecule 1 (STIM1), which resides on the ER/SR membrane and in response to Ca^2+^ depletion aggregates to ER/SR locales proximal to the plasma membrane [220]. STIM1 interacts with the plasma membrane channel Orai Ca^2+^ release-activated Ca^2+^ modulator 1 (Orai1) [221], which facilitates extracellular Ca^2+^ entry into the ER/SR. Indeed, the importance of SOCE for skeletal muscle health is apparent in mice with STIM1 haploinsufficiency, which exhibit markedly greater muscle fatigability [222]. Furthermore, mice that express a dominant negative form of Orai1 display both reduced skeletal muscle mass as well as enhanced susceptibility to fatigue during repeated muscle contraction [223]. Investigations regarding the relevance of SOCE to aging-associated functional deficits have revealed that skeletal muscle SOCE function is reduced in aged animals despite sustained STIM1 and Orai1 mRNA expression [224,225], although this has been challenged [226]. During *ex vivo* contractility assays, it was revealed that inhibition of SOCE reduces contractile activity in skeletal muscle from young but not aged animals, an effect that was most prominent at high frequency stimulation [225]. These data suggest that there exists a lack of SOCE contribution to contractile function at high intensities in geriatric muscle. Interestingly, aged skeletal muscle was shown to contain reduced abundance of the synaptophysin-related membrane protein Mitsugumin 29 (MG29) that regulates SR and transverse tubule contact sites [227]. A lack of MG29 has been shown to reduce skeletal muscle contractile function [228] as well as compromise SOCE [224]. These findings reveal that in addition to enhanced Ca^2+^ leakage from the ER/SR, inadequate SOCE may contribute to a dysregulation of Ca^2+^ homeostasis that compromises function in aged skeletal muscle.

## 4. Therapeutics for the Treatment of Sarcopenia

The evidence for non-pharmacological approaches to mitigating or preventing sarcopenia suggests that both exercise [229,230] and to a lesser extent caloric restriction [231,232,233] reduce age-associated muscle quality decline, although there exist concerns that a hypocaloric diet may also reduce lean mass [234,235]. Furthermore, although combined caloric restriction and exercise may yield the most benefit, attrition to lifestyle intervention may not be a practical solution for some individuals suffering from comorbidities or mobility impairments. Thus, pharmacological agents that mimic the benefits gained from routine exercise, caloric restriction, or dietary interventions are attractive therapeutic targets to prevent or mitigate sarcopenia (Figure 4).

### 4.1. Branched-Chain Amino Acid Supplementation

Branched-chain amino acids (BCAAs) play a crucial role as both signaling molecules and substrates for protein synthesis. Low blood levels of BCAAs are associated with sarcopenia and reduced physical function [236]. Ingestion of BCAAs was shown to enhance early post prandial myofibrillar protein synthesis in older males (~70 years old) [237]. Among the three BCAAs (leucine, valine, and isoleucine), leucine is considered to be the primary modulator of muscle anabolism. Analysis of dietary leucine intake and muscle mass have revealed that leucine intake is correlated to the preservation of lean mass over the course of 6 years in older (>65 years) but not younger subjects [238]. Additionally, Devries et al. [239] found that the acute induction of MPS in healthy older women (65–75 years) that ingested either 10 g of milk or 25 g of whey protein isolate was similar when leucine content was matched. These data reveal that the BCAA content of a meal is a primary determinant of the anabolic response to feeding. However, it is necessary to keep in mind that the consumption of BCAAs without dietary inclusion of other essential amino acids may not be as beneficial, as protein synthesis requires both anabolic signaling and adequate availability of other amino acids [240]. Nonetheless, short term supplementation of BCAAs in combination with other amino acids for 5 weeks revealed positive effects on sarcopenic parameters in elderly patients. However, the reported benefits were lost after 12 weeks of discontinuation [241]. Taken together, the available literature suggests that greater BCAA consumption, particularly that of leucine, in combination with adequate protein intake may be effective at preserving muscle mass in elderly individuals.

### 4.2. Rapamycin, Rapalogs, and Calorie Restriction

Recently, interest has grown over the use of mTORC1 inhibitors and calorie restriction mimetics as a potential treatment for sarcopenia. Indeed, the use of rapamycin and rapalogs have proven to be beneficial in ameliorating age-associated skeletal muscle defects in preclinical models [43,117,122]. Interestingly, while rapamycin is considered to act through a similar mechanism as calorie restriction, a recent study by Ham et al. [125] has revealed that rapamycin improves aging-associated skeletal muscle outcomes through mechanisms that are separate from caloric restriction. Rapamycin treatment tended to reverse age related gene expression patterns which were often augmented by caloric restriction in old animals, indicating divergent regulation of gene expression between the two interventions. Caloric restriction has been shown to reduce muscle fiber central nucleation and p62 protein accumulation in skeletal muscle TSC1-knockout mice, without altering Sqstm1 gene expression, autophagic flux, or diminishing mTORC1 activation [125]. These data indicate that caloric restriction can improve muscle degeneration through mechanisms that are independent of mTORC1 suppression. However, it is worth noting that improvements in skeletal muscle mass after caloric restriction were largely only positive when muscle mass was normalized to body mass, which tended to be significantly lower in aged calorie restricted mice. After caloric restriction, absolute mass of several hindlimb muscles in old mice were the same or diminished when compared to ad libitum fed old mice, which was also mirrored in fiber cross sectional area [125]. Furthermore, absolute force tended to be similar or lower in aged calorie restricted mice when compared to aged ad libitum mice, while specific force tended to improve [125]. These results suggest that while calorie restriction may negatively affect absolute muscle mass, relative muscle mass and quality may be improved during aging (Figure 4).

### 4.3. Mitochondrial Uncouplers

Indeed, mTORC1 inhibitors are not the only pharmaceutical agents with alleged benefits for aging associated muscle defects. Dietary supplementation with the mitochondrial uncoupler BAM15 induced markers of mitophagy and improved muscle mass in a murine model of sarcopenic obesity [215]. Administration of BAM15 reduced both ER stress and apoptotic signaling in the skeletal muscle of sarcopenic obese mice. Interestingly, while mTORC1 activation was not diminished there was a marked upregulation of AMPK phosphorylation concomitant with greater mitochondrial volume. Similar to BAM15, *in vitro* administration of urolithin A to the myoblast line of C2C12 cells reduces Δψm and enhances mitophagy. When given to aged mice on either a chow or a high-fat diet, dietary administration of urolithin A potently enhanced skeletal muscle physical function without changes to body mass. Improvements in physical function were concomitant with enhanced AMPK phosphorylation and greater abundance of markers of autophagy and mitophagy [242]. These data provide evidence that enhancing mitophagy to maintain the health of the skeletal muscle mitochondrial pool is advantageous for treatment of sarcopenic obesity (Figure 4).

### 4.4. Androgens and SARMs

Given the association between sarcopenia and hypogonadism, targeting of the androgen receptor via selective androgen receptor modulators (SARMs) and testosterone has been investigated as viable therapeutics for sarcopenia. The administration of exogenous testosterone, while known to substantially enhance skeletal muscle mass [243], is also associated with adverse outcomes regarding prostate health [244]. Co-administration of testosterone and the type II 5α-reductase inhibitor finasteride to older men (60–80 yrs. old) results in significant enhancements in skeletal muscle mass as well as strength without adverse outcomes in prostate size, indicating that androgen receptor activation in combination with type II 5α-reductase inhibitors may be a viable strategy to improve sarcopenia [245]. Activation of the androgen receptor using nonsteroidal SARMs has several benefits over testosterone due to their tissue selective anabolic properties. In female patients with sarcopenia, administration of the SARM MK-0773 for 6 months produced significant enhancements in lean body mass but failed to significantly improve physical function over placebo [246]. In contrast, the SARM GTx-024 improved both lean body mass and functional outcomes in healthy older men and women, which was concomitant with favorable metabolic indices (Figure 4).

### 4.5. Myostatin Inhibitors

Notably, there have also been several drug candidates produced to target myostatin, a potent negative regulator of skeletal muscle mass. Administration of the human monoclonal antibody REGN1033 to target myostatin in mice enhanced both muscle mass and force production and prevented muscle atrophy induced by limb immobilization. Treatment of aged mice with REGN1033 led to significantly increased muscle mass and force production without a loss in specific force, indicating a preservation of muscle quality [247]. Subcutaneous injections of the anti-myostatin antibody LY2495655 were tested in humans aged 75 and older who had a fall within the previous year and exhibited low muscle strength to determine how LY2495655 would alter appendicular lean mass and physical function [248]. The results indicated that LY2495655 treatment improved appendicular lean mass and physical performance measures as objective measures that heavily relied on power generation. Like antibodies that bind and neutralize myostatin, monoclonal antibodies that bind type II activin receptors and thus prevent myostatin associated signaling have also shown promise. For example, in a 24-week, randomized, double-blind, placebo-controlled study, the use of the type II activin receptor antagonist bimagrumab in sarcopenic community-living men and women aged 65 and older resulted in a greater accrual of lean body mass and appendicular lean mass than placebo [249]. The greater accrual of lean body mass and muscle volume were evident in as short as two weeks of treatment [249]. Additionally, while bimagrumab treatment did not improve six-minute walk distance between groups, there was a significant improvement in six-minute walk distance in the slowest walkers after bimagrumab treatment when compared to placebo controls. This suggests that bimagrumab treatment improved functional outcomes in those with the largest deficits in mobility based physical function. Additional work is needed to optimize bimagrumab, or other compounds that block type II activin receptors and evaluate muscle function, mobility, and muscle mass after long term treatment in the elderly. However, the initial data after acute administration suggest that this might be a beneficial approach for reducing sarcopenia (Figure 4).

### 4.6. Present Challenges and Future Direction

Currently, there are no approved drugs for the treatment of sarcopenia. Recent and past findings have shown that a finely tuned balance between anabolic and catabolic processes is required to maintain adequate muscle quality and mass in older ages. Questions remain regarding the appropriate degree of caloric restriction to combat age related muscle dysfunction. An important variable to consider when interpreting the impact of caloric restriction is to differentiate between the benefit of an energy deficit vs. the maintenance of a reduced body weight. A perpetual caloric deficit would result in a chronic loss of body mass and exacerbate sarcopenia. Thus, organisms adapt to reduced caloric intake via weight loss and energy sparing processes to reach an equilibrium between calorie expenditure and calorie intake. It is unclear how much of the purported benefit of calorie restriction is due to an incurred energy deficit vs. the maintenance of a healthier lower body weight (primarily reduced fat mass). Additionally, caloric restriction is difficult to execute consistently and effectively in a clinical setting. It is possible that regimens consisting of interspersed periods of caloric restriction with isocaloric diets may activate cellular pathways that confer protection against sarcopenia without compromising absolute skeletal muscle mass. Furthermore, while extensive lifestyle modification may not be a practical and viable option for elderly patients, small amounts of physical activity may improve autophagy/mitophagy and be augmented pharmaceutically to allow for even a greater benefit on muscle quality. A combination of exercise that is coupled with supplementation with the polyphenol resveratrol has also been shown to augment exercise adaptations in older men and women to offset sarcopenia [250]. This suggests that the benefit incurred from physical activity can be pharmaceutically amplified. Future research should elucidate interactions between pharmaceutical agents with exercise and/or dietary manipulation as potential avenues to combat sarcopenia.

## 5. Conclusions

The primary cause of cellular dysregulation that contributes to sarcopenia development remains elusive. However, recent findings implicate dysregulated proteostasis as an important characteristic of skeletal muscle in aged hosts that may contribute to muscle atrophy and functional decline leading to sarcopenia. Additionally, given the multi-factorial aspect of sarcopenia, effective treatment will likely be best achieved by addressing multiple pathways that contribute to the loss of muscle function and mass with aging. Furthering our understanding of the causes and consequence of dysfunctional proteostasis in aged skeletal muscle will allow for the development of targeted treatments to ameliorate one of the most debilitating geriatric conditions.

## Figures and Tables

**Figure 1 cells-12-00249-f001:**
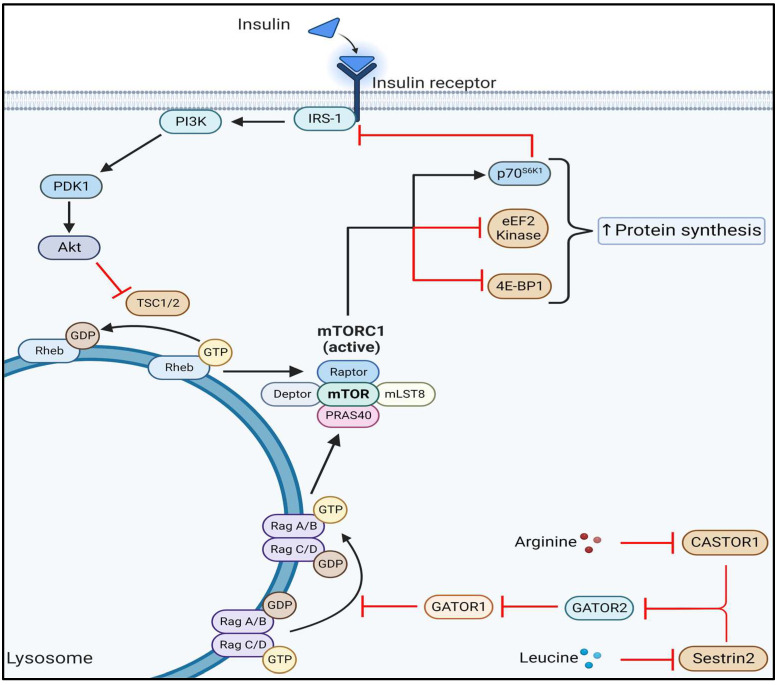
**Activation of mTORC1 by growth factors and amino acids.** Activation of mTORC1 requires the integration of extracellular and intracellular signals. Binding of insulin to the insulin receptor induces a cascade of signaling events that culminate in Akt-mediated inhibition of TSC1/2. Phosphorylation of TSC1/2 by Akt prevents TSC1/2 GAP activity towards RHEB, allowing for activation of mTORC1. In response to amino acid availability, CASTOR1 and Sestrin2 inhibition of GATOR2 is relieved, allowing for GATOR2 inhibition of GATOR1. Inhibition of GATOR1 allows for Rag-mediated activation of mTORC1. Phosphorylation of p70^S6K1^, 4e-BP1- and eEF2 kinase by mTORC1 enhances protein translation initiation and elongation. Activated p70^S6K1^ can exert negative feedback on Akt via inhibition of the insulin signaling cascade.

**Figure 2 cells-12-00249-f002:**
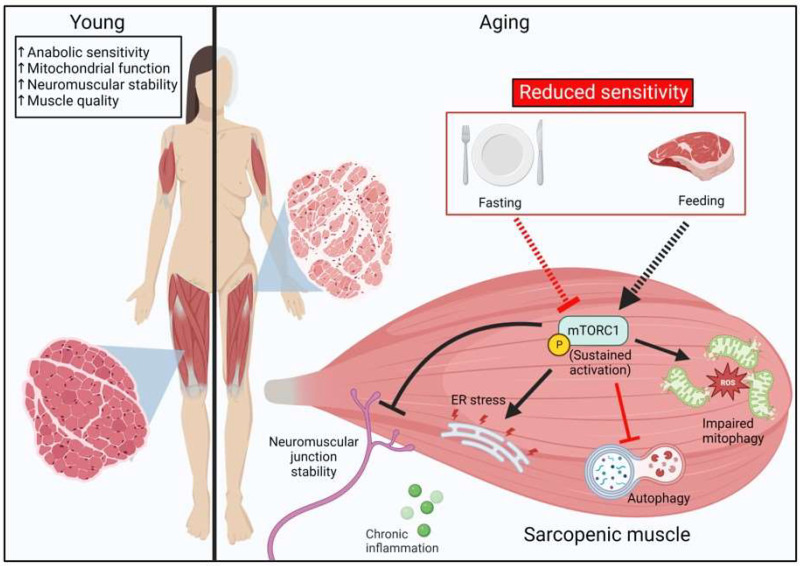
**Proposed model of mTORC1-mediated muscle atrophy in sarcopenia.** Aged skeletal muscle exhibits anabolic resistance to hyperaminoacidemia characterized by an impaired protein synthetic response. Basal mTORC1 exhibits sustained activation in aged skeletal muscle, that can contribute to neuromuscular junction instability, ER stress, and impaired mitophagy. Poor cellular quality control mechanisms result in a compromised mitochondrial pool, leading to impaired bioenergetics and enhanced reactive oxygen species production.

**Figure 3 cells-12-00249-f003:**
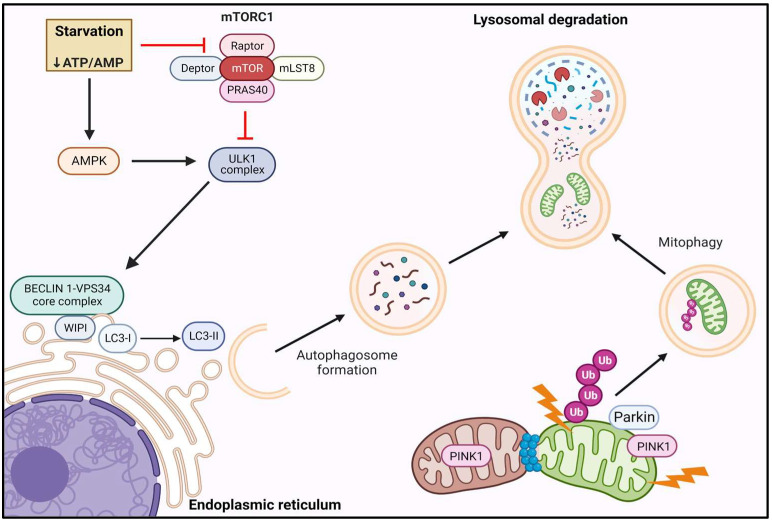
**Autophagy and mitophagy pathways.** Autophagy induction is negatively regulated by mTORC1 signaling through direct inhibition of ULK1 activity. During periods of nutrient deprivation, mTORC1 activity is blunted, thereby releasing inhibition of the ULK1 initiation complex. Additionally, activation of AMPK under conditions of energetic stress directly facilitates ULK1 activation. Recruitment of the Beclin-VPS34 core complex to the ER facilitates LC3 lipidation and autophagosome membrane expansion. Loss of mitochondrial Δψm triggers OMM PINK1 accumulation and Parkin recruitment, which results in polyubiquitination and sequestration of damaged mitochondria in autophagosomes for lysosomal degradation.

**Figure 4 cells-12-00249-f004:**
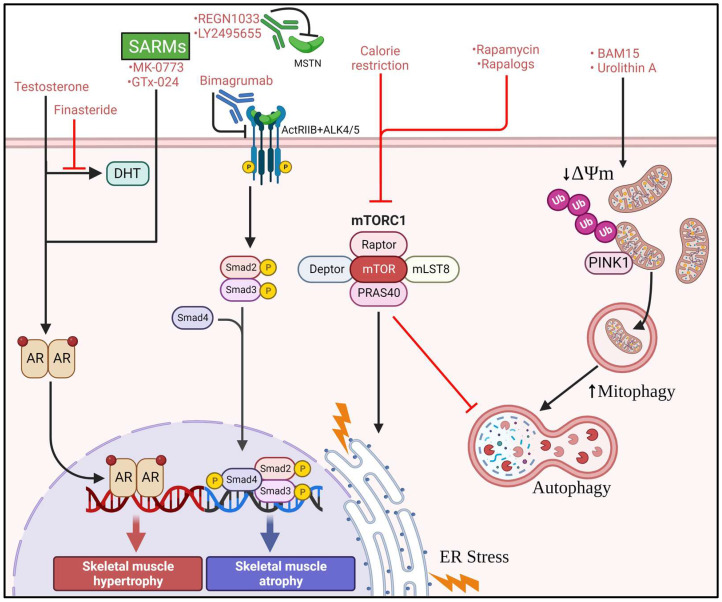
**Potential therapeutic interventions to treat sarcopenia.** Calorie restriction, rapamycin, and rapalogs can enhance skeletal muscle quality through inhibition of mTORC1 signaling and enhancement of autophagy and cellular quality control mechanisms. Administration of BAM15 and urolithin A improves muscle mass and quality in sarcopenic obesity through depolarization of mitochondria and subsequent mitophagic clearance, thereby dynamically maintaining the health of the mitochondrial pool. Testosterone and SARMs improve skeletal muscle mass through androgen receptor (AR)-mediated mechanisms, while finasteride is used to diminish androgenic signaling by inhibiting dihydrotestosterone (DHT) production. Antibodies used to target both myostatin (MSTN) or the activin type II receptor inhibit myostatin-mediated signaling upstream of SMAD2/3, thereby reducing catabolic gene expression and facilitating skeletal muscle anabolism.

## Data Availability

The data presented and reviewed in this manuscript are openly available in published forms as documented in the references of this manuscript.

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
