# Peer review of "Age-Related Dysfunction in Proteostasis and Cellular Quality Control in the Development of Sarcopenia"

_cells, 2023, doi:10.3390/cells12020249_

Round 1

Reviewer 1 Report

Paez et al. presented a review with the title „Age-related dysfunction in proteostasis and cellular quality control in the development of sarcopenia”. The authors summarise  the recent advances of the molecular mechanisms driving sarcopenia and describe potential therapeutic interventions to counteract it. The review was clear, and provides us with a concise description of sarcopenia, the key mechanisms, and the possible therapies. There are comprehensive reviews written already on this topic, so the novelty of this work is not high, however, this includes the latest observations as well. All in all, this review has some new aspects that make it valuable.

Some improvements could be made:

1) A paragraph describing the role of calcium ions in sarcopenia should be added. There are several studies demonstrating the presence of a calcium overload and a reduction of the soce mechanism in the sarcopenic muscle, with all the ensuing consequences

2) Among the possible therapies for the treatment of sarcopenia, there is no mention of the use of branched-chain amino acids. In the literature, however, there are recent studies that demonstrate their beneficial effect on the sarcopenic muscle

Author Response

RESPONSE TO THE REVIEWER  

We wish to thank the Reviewers for their helpful comments. We believe that their suggestions have strengthened and clarified the manuscript.  

Response to Reviewer 1 Comments 

Reviewer 1 Comments Paez et al. presented a review with the title “Age-related dysfunction in proteostasis and cellular quality control in the development of sarcopenia”. The authors summarise  the recent advances of the molecular mechanisms driving sarcopenia and describe potential therapeutic interventions to counteract it. The review was clear, and provides us with a concise description of sarcopenia, the key mechanisms, and the possible therapies. There are comprehensive reviews written already on this topic, so the novelty of this work is not high, however, this includes the latest observations as well. All in all, this review has some new aspects that make it valuable.  

Authors comments: Thank you for your comments and suggestions to strengthen our manuscript.  

Reviewer 1 Comments Some improvements could be made:  

1) A paragraph describing the role of calcium ions in sarcopenia should be added. There are several studies demonstrating the presence of a calcium overload and a reduction of the soce mechanism in the sarcopenic muscle, with all the ensuing consequences  

Authors comments: We appreciate the insight  and recommendation from the Reviewer. We have updated the manuscript to include a paragraph describing recent literature regarding SR calcium leakage and contractile function in aged skeletal muscle.  

Reviewer 1 Comments 2) Among the possible therapies for the treatment of sarcopenia, there is no mention of the use of branched-chain amino acids. In the literature, however, there are recent studies that demonstrate their beneficial effect on the sarcopenic muscle.  

Authors comments:  Thank you for your suggestion. A section regarding the use of BCAA’s and their beneficial effects on sarcopenia have been added.  

Reviewer 2 Report

Comments on cells-2121359

In this study, the author has studied “Age-related dysfunction in proteostasis and cellular quality control in the development of sarcopenia.” A lot of studies have already been carried out on a similar topic, and comprehensive data is available in the literature. The English language used in the manuscript needs major improvements as there are some punctuation and grammatical mistakes present throughout the manuscript. Experimental designs required more clarity. Moreover, research models are not discussed in an understandable manner. Repetition of lines is common, which reflects that the author needs a more comprehensive way of thinking. The manuscript is too long but without any specific illustrations. It is strongly recommended to add figures where is applicable to catch the eyes of the readers.

Specific comments:

1.      The Abstract needs to be critically revised, please add some more information to catch the eyes of the readers.

2.      Please add more strong keywords.

3.      Page 1, line 32: “development of age-related diseases and disability….” Please add a reference here.

4.      Page 2, line 54: “blunted recovery after insult…” Please revise it.

5.      Page 2: The whole introduction section looks general. Authors are advised to revise the introduction section carefully and add relevant data to support the problem statement and make a connection between each paragraph. There is no such information between proteostasis and cellular quality control in the development of sarcopenia rather than the causes of sarcopenia. The authors only discussed general information about sarcopenia. Overall, an introduction needs a major revision.

6.      Page 2: What is the research gap and novelty of the present review?

7.      Page 3, line 80-81: “Given mTORC1 initiates a resource…” Please remove the word ‘given.’

8.      “mTOR regulation of muscle protein synthesis” Please add a signaling figure in this section

9.      Please add more information in section 2.1.3. and 2.1.4 with proper illustrations.

10.  Page 5, line 209-210: “Hyperaminoacidemia is not the only stimulus that senescent muscle displays a blunted anabolic response to;” Please revise the sentence.

11.  Page 7, line 300-301: “muscle specific RAPTOR knockout also produced skeletal muscle atrophy.” Please add the full name of the term when used for the first time.

12.  Please also insert/place Figure 1 in the text.

13.  “Autophagy and mitophagy” Please add a figure in this heading.

14.  Page 11, line 487-488: “Nonetheless, it is intriguing that two effective interventions for mitigation of sarcopenia and aging, namely caloric restriction [213] and exercise [183,210,214,215]…” It is not appropriate to add many references in one place.

15.  The purpose of sections 3.3 and 3.4 looks similar. Please revise the headings.

16.  Please also insert/place Figure 2 in the text.

17.  It is recommended to add a new heading “Present challenges and future direction.”

18.  Authors are advised to proofread the manuscript to overcome grammatical mistakes.’

19.  Authors are advised to revise headings and subheadings. There are some places where authors add a full-stop (.) in headings, please revise them.

20.  Most of the references are outdated; please revise them and add updated data.

21.  It is recommended to add a list of abbreviations.

Author Response

RESPONSE TO THE REVIEWERS

We wish to thank the Reviewers for their helpful comments. We believe that their suggestions have strengthened and clarified the manuscript.

Response to Reviewer 1 Comments

Reviewer 1 Comments Paez et al. presented a review with the title “Age-related dysfunction in proteostasis and cellular quality control in the development of sarcopenia”. The authors summarise  the recent advances of the molecular mechanisms driving sarcopenia and describe potential therapeutic interventions to counteract it. The review was clear, and provides us with a concise description of sarcopenia, the key mechanisms, and the possible therapies. There are comprehensive reviews written already on this topic, so the novelty of this work is not high, however, this includes the latest observations as well. All in all, this review has some new aspects that make it valuable.

Reviewer 2 Comments

In this study, the author has studied “Age-related dysfunction in proteostasis and cellular quality control in the development of sarcopenia.” A lot of studies have already been carried out on a similar topic, and comprehensive data is available in the literature. The English language used in the manuscript needs major improvements as there are some punctuation and grammatical mistakes present throughout the manuscript. Experimental designs required more clarity. Moreover, research models are not discussed in an understandable manner. Repetition of lines is common, which reflects that the author needs a more comprehensive way of thinking. The manuscript is too long but without any specific illustrations. It is strongly recommended to add figures where is applicable to catch the eyes of the readers.

Authors comments: Thank you for your comments and suggestions to strengthen our manuscript. We have made extensive revisions and rewrote the manuscript. We have shortened parts of the manuscript, but the other Reviewer asked us to add a section to the paper so the overall length is similar to the maximum allowed for the review. We have added new figures as you have suggested.

Reviewer 2 Comments Specific comments:

  1. The Abstract needs to be critically revised, please add some more information to catch the eyes of the readers.

Authors commentsThe abstract has been updated with more information and maintained at the ~200 word abstract limit as required by the journal. Unfortunately, this does not provide much space for adding a lot of information. We have revised the abstract as follows:

Abstract: Sarcopenia is a debilitating skeletal muscle disease that appears in the last decades of life and is characterized by marked deficits in muscle strength, mass, and metabolic health. The multifactorial causes of sarcopenia have proven difficult to treat and involve a complex interplay between environmental factors and intrinsic ageassociated changes. It is generally accepted that sarcopenia results in a progressive loss of skeletal muscle function that exceeds the loss of mass, indicating that while loss of muscle mass is important, loss of muscle quality is the primary defect with advanced age. Furthermore, preclinical models have suggested that aged skeletal muscle exhibits defects in cellular quality control such as the degradation of damaged mitochondria.  Recent evidence suggests that a dysregulation of proteostasis, an important regulator of cellular quality control, is a significant contributor to agingassociated declines in muscle quality, function, and mass. Although mTORC1 plays a critical role in cellular control, including skeletal muscle hypertrophy, paradoxically, sustained activation of skeletal muscle mTORC1 recapitulates several characteristics of sarcopenia. Pharmaceutical inhibition of mTORC1 as well as ca-loric restriction significantly improves muscle quality in aged animals, however, the mechanisms controlling cellular proteostasis are not fully known. This information is important if we are to develop effective therapeutic strategies to mitigate or prevent sarcopenia and associated disability. This review identifies recent advances in unraveling the molecular mechanisms of proteostatis driving age-associated muscle loss and suggests potential therapeutic interventions to slow or prevent sarcopenia.

  1. Reviewer 2 Comments Please add more strong keywords.

Authors comments: The following has been added/revised: Keywords: sarcopenia; skeletal muscle; atrophy; mitochondria; autophagy; mitophagy; aging; mTORC1; dynapenia; caloric restriction; muscle protein synthesis; ubiquitin proteasome; anabolic resistance; rapamycin

  1. Reviewer 2 Comments Page 1, line 32: “development of age-related diseases and disability….” Please add a reference here.

Authors comments: References have been added to this section as requested.

  1. Page 2, line 54: “blunted recovery after insult…” Please revise it.

Authors comments: This sentence was revised and edited.

  1. Reviewer 2 Comments Page 2: The whole introduction section looks general. Authors are advised to revise the introduction section carefully and add relevant data to support the problem statement and make a connection between each paragraph. There is no such information between proteostasis and cellular quality control in the development of sarcopenia rather than the causes of sarcopenia. The authors only discussed general information about sarcopenia. Overall, an introduction needs a major revision.

Authors comments: Thank you for your suggestions. The introduction has been edited to include a better transition between each paragraph and additional content has been added to highlight proteostasis as critical and ongoing area of research for the development of sarcopenia.

  1. Reviewer 2 Comments Page 2: What is the research gap and novelty of the present review?

Authors comments: The present review aims to summarize previous and recent findings regarding the roles of protein synthesis, degradation, and cellular quality control mechanisms in sarcopenic muscle. While there exists similar reviews on these individual topics, we incorporate novel and recent studies regarding the interplay between anabolic and catabolic pathways while giving the reader historical context with seminal findings. This allows the reader to properly contextualize and juxtapose recent findings within the broader scope of the literature.

Furthermore, we critically discuss the efficacy of caloric restriction as a treatment for sarcopenia and highlight literature that challenges conventions regarding the putative contributors to sarcopenia.

  1. Reviewer 2 Comments Page 3, line 80-81: “Given mTORC1 initiates a resource…” Please remove the word ‘given.’

Authors comments: “given” was removed.

  1. Reviewer 2 Comments “mTOR regulation of muscle protein synthesis” Please add a signaling figure in this section

Authors comments: We have added a new figure as suggested by the Reviewer.

  1. Reviewer 2 Comments Please add more information in section 2.1.3. and 2.1.4 with proper illustrations.

Authors comments: Section 2.1.3 was removed from the manuscript to reduce the length of the manuscript at the request of the reviewer. Section 2.1.4 was merged as a continuation of mTORC1 signaling in response to feeding and a figure has been added.

  1. Reviewer 2 Comments Page 5, line 209-210: “Hyperaminoacidemia is not the only stimulus that senescent muscle displays a blunted anabolic response to;” Please revise the sentence.

Authors comments: This section was removed from the manuscript to reduce the length of the manuscript at the request of the reviewer.

  1. Reviewer 2 Comments Page 7, line 300-301: “muscle specific RAPTOR knockout also produced skeletal muscle atrophy.” Please add the full name of the term when used for the first time.

Authors comments: The full name was added and abbreviations within the text have been updated.

  1. Reviewer 2 Comments Please also insert/place Figure 1 in the text.

Authors comments: Figure 1 has been referenced in the text as suggested. The other figures were also referenced in the appropriate texts

  1. Reviewer 2 Comments “Autophagy and mitophagy” Please add a figure in this heading.

Authors comments: A figure has been added to highlight these pathways.

  1. Reviewer 2 Comments Page 11, line 487-488: “Nonetheless, it is intriguing that two effective interventions for mitigation of sarcopenia and aging, namely caloric restriction [213] and exercise [183,210,214,215]…” It is not appropriate to add many references in one place.

Authors comments: The number of references in this sentence were reduced.

  1. Reviewer 2 Comments The purpose of sections 3.3 and 3.4 looks similar. Please revise the headings.

Authors comments:These sections have been reorganized and the section headings have been revised.

  1. Reviewer 2 Comments Please also insert/place Figure 2 in the text

Authors comments: We have referenced Figure 2 in the text.

  1. Reviewer 2 Comments It is recommended to add a new heading “Present challenges and future

direction.”

Authors comments: A new heading for present challenges and future directions has been added.

  1. Reviewer 2 Comments Authors are advised to proofread the manuscript to overcome grammatical mistakes.’

Authors comments:The manuscript was reviewed by the authors and edited for grammatical mistakes.

  1. Reviewer 2 Comments Authors are advised to revise headings and subheadings. There are some places where authors add a full-stop (.) in headings, please revise them.

Authors comments: The headings have been revised as suggested.

  1. Reviewer 2 Comments Most of the references are outdated; please revise them and add updated data.

Authors comments:  Several older references have been removed and new references have been added in. References of landmark studies and seminal findings have been retained in the manuscript.

  1. Reviewer 2 Comments It is recommended to add a list of abbreviations.

Authors comments: An abbreviations section has been added to the end of the text.

Round 2

Reviewer 1 Report

This second version of the manuscript is improved. However, it needs some minor revisions.

In paragraph 3.2.4.  I suggest adding a few lines on the SOCE mechanism, which cannot be excluded when talking about calcium homeostasis. It is known that SOCE is severely reduced in sarcopenic muscle without the presence of an alteration of STIM1/Orai1 mRNA levels and the decline of the SOCE mechanism may contribute to the reduction of muscle strength and the increase of fatigue susceptibility. doi.org/10.1111/j.1474-9726.2008.00408.x; doi: 10.3390/cells10102722; doi: 10.18632/aging.100335; 

In line 732 reference 247 doesn't exist. Please replace it with the correct reference.

Author Response

Thank you for your supportive comments and your suggestions for improvements to our paper.

Reviewer 1 Comments. This second version of the manuscript is improved. However, it needs some minor revisions.

Author responses: Thank you for your comments and your suggestions for further improvements to our paper.

Reviewer 1 Comments. In paragraph 3.2.4.  I suggest adding a few lines on the SOCE mechanism, which cannot be excluded when talking about calcium homeostasis. It is known that SOCE is severely reduced in sarcopenic muscle without the presence of an alteration of STIM1/Orai1 mRNA levels and the decline of the SOCE mechanism may contribute to the reduction of muscle strength and the increase of fatigue susceptibility. doi.org/10.1111/j.1474-9726.2008.00408.x; doi: 10.3390/cells10102722; doi: 10.18632/aging.100335; 

Author responses. We have now added the following to address the issue of SOCE. We have also uploaded the red font paper showing the insertion below in the previous version of the manuscript. 

In addition to dysfunctional Ca2+ reuptake, there is evidence that store-operated Ca2+ entry (SOCE) is inefficient in sarcopenic muscle and may contribute to declines in skeletal muscle performance. SOCE is critical to maintain intracellular Ca2+ homeostasis as ER/SR Ca2+ stores are depleted. In brief, reductions in ER/SR Ca2+ are sensed by the Ca2+ sensor stromal interaction molecule 1 (STIM1), which resides on the ER/SR membrane and in response to Ca2+ depletion aggregates to ER/SR locales proximal to the plasma membrane [221]. STIM1 interacts with the plasma membrane channel Orai Ca2+ release-activated Ca2+modulator 1 (Orai1) [222], which facilitates extracellular Ca2+ entry into the ER/SR. Indeed, the importance of SOCE for skeletal muscle health is apparent in mice with STIM1 haploinsufficiency, which exhibit markedly greater muscle fatigability [223]. Furthermore, mice that express a dominant negative form of Orai1 display both reduced skeletal muscle mass as well as enhanced susceptibility to fatigue during repeated muscle contraction [224]. Investigations regarding the relevance of SOCE to aging-associated functional deficits have revealed that skeletal muscle SOCE function is reduced in aged animals despite sustained STIM1 and Orai1 mRNA expression [225,226], although this has been challenged [227]. During ex vivo contractility assays, it was revealed that inhibition of SOCE reduces contractile activity in skeletal muscle from young but not aged animals, an effect that was most prominent at high frequency stimulation [226]. These data suggest that there exists a lack of SOCE contribution to contractile function at high intensities in geriatric muscle. Interestingly, aged skeletal muscle was shown to contain reduced abundance of the synaptophysin-related membrane protein Mitsugumin 29 (MG29) that regulates SR and transverse tubule contact sites [228]. A lack of MG29 has been shown to reduce skeletal muscle contractile function [229] as well as compromise SOCE [225]. These findings reveal that in addition to enhanced Ca2+ leakage from the ER/SR, inadequate SOCE may contribute to a dysregulation of Ca2+ homeostasis that compromises function in aged skeletal muscle.

Reviewer 1 Comments. In line 767 reference 247 doesn't exist. Please replace it with the correct reference

Author responses: Thank you for noting this problem. We have reinserted the reference and we believe that the manuscript is now referenced correctly.

Reviewer 2 Report

The authors have carefully addressed all the comments. So, the manuscript should be accepted in its present form.

Author Response

Thank you for your review and your suggestions for improving our manuscript. We appreciate your support for accepting the manuscript. A couple of minor issues were corrected that included reformatting the references. Reviewer 1 also asked us to add information to discuss the SOCE mechanism of calcium regulation. The revised manuscript in its clean form is uploaded, but the red font version of the addition of the SOCE information to the previous version of the manuscript is also attached to the non-published section of this submission.

The new addition is copied below:

In addition to dysfunctional Ca2+ reuptake, there is evidence that store-operated Ca2+ entry (SOCE) is inefficient in sarcopenic muscle and may contribute to declines in skeletal muscle performance. SOCE is critical to maintain intracellular Ca2+ homeostasis as ER/SR Ca2+ stores are depleted. In brief, reductions in ER/SR Ca2+ are sensed by the Ca2+ sensor stromal interaction molecule 1 (STIM1), which resides on the ER/SR membrane and in response to Ca2+ depletion aggregates to ER/SR locales proximal to the plasma membrane [221]. STIM1 interacts with the plasma membrane channel Orai Ca2+ release-activated Ca2+modulator 1 (Orai1) [222], which facilitates extracellular Ca2+ entry into the ER/SR. Indeed, the importance of SOCE for skeletal muscle health is apparent in mice with STIM1 haploinsufficiency, which exhibit markedly greater muscle fatigability [223]. Furthermore, mice that express a dominant negative form of Orai1 display both reduced skeletal muscle mass as well as enhanced susceptibility to fatigue during repeated muscle contraction [224]. Investigations regarding the relevance of SOCE to aging-associated functional deficits have revealed that skeletal muscle SOCE function is reduced in aged animals despite sustained STIM1 and Orai1 mRNA expression [225,226], although this has been challenged [227]. During ex vivo contractility assays, it was revealed that inhibition of SOCE reduces contractile activity in skeletal muscle from young but not aged animals, an effect that was most prominent at high frequency stimulation [226]. These data suggest that there exists a lack of SOCE contribution to contractile function at high intensities in geriatric muscle. Interestingly, aged skeletal muscle was shown to contain reduced abundance of the synaptophysin-related membrane protein Mitsugumin 29 (MG29) that regulates SR and transverse tubule contact sites [228]. A lack of MG29 has been shown to reduce skeletal muscle contractile function [229] as well as compromise SOCE [225]. These findings reveal that in addition to enhanced Ca2+ leakage from the ER/SR, inadequate SOCE may contribute to a dysregulation of Ca2+ homeostasis that compromises function in aged skeletal muscle.
